# Chronic Bronchitis Affects Outcomes in Smokers without Chronic Obstructive Pulmonary Disease (COPD)

**DOI:** 10.3390/jcm11164886

**Published:** 2022-08-20

**Authors:** Alvise Casara, Graziella Turato, Marta Marin-Oto, Umberto Semenzato, Davide Biondini, Mariaenrica Tinè, Nicol Bernardinello, Elisabetta Cocconcelli, Pablo Cubero, Elisabetta Balestro, Paolo Spagnolo, Josè M. Marin, Manuel G. Cosio, Marina Saetta, Erica Bazzan

**Affiliations:** 1Department of Cardiac, Thoracic, Vascular Sciences and Public Health, University of Padova, 35128 Padova, Italy; 2Respiratory Service, Hospital Clinico Universitario, 50009 Zaragoza, Spain; 3Translational Research Unit (IIS Aragón), Hospital Universitario Miguel Servet, 50009 Zaragoza, Spain; 4Department of Medicine, University of Zaragoza School of Medicine, 50009 Zaragoza, Spain; 5Meakins-Christie Laboratories, Respiratory Division, McGill University, Montreal, QU 000004, Canada

**Keywords:** chronic bronchitis, COPD, smoking

## Abstract

Background. Chronic bronchitis (CB) importantly affects outcomes in smokers with COPD, but the effects on smokers without COPD are less well known and less emphasized. The aim of our study was to investigate the possible effects of CB on clinical outcomes in smokers without COPD (noCOPD) and compare them with the effects in smokers with COPD (COPD). Methods. For that purpose, we studied 511 smokers, 302 with and 209 without COPD, followed for 10 years in an academic COPD ambulatory setting. Chronic bronchitis was defined as the presence of cough and sputum production for at least 3 months in each of two consecutive years. All subjects underwent clinical and functional examination with spirometry, diffusion capacity (DLco), 6-min walking test (6MWT), mMRC Dyspnoea Scale, COPD Assessment Test (CAT), and recording of annual frequency of exacerbations. All-cause mortality during follow-up was recorded. Results. 27% of noCOPD and 45% of COPD had CB. noCOPD with CB had lower FEV_1_ and DLco, worse 6MWT, more dyspnoea, a higher number of exacerbations and lower survival than noCOPD without CB. CB did not affect FEV_1_ decline in noCOPD but it significantly did in COPD. Conclusions. The presence of chronic bronchitis in smokers without COPD will significantly affect symptoms, quality of life, and survival, underlining the importance of recognizing the condition and managing it accordingly.

## 1. Introduction

Chronic cough and sputum production are among the most commonly reported medical symptoms of which chronic bronchitis (CB), a clinical condition originally defined as “chronic” “recurrent” increase in the volume of mucoid bronchial secretions sufficient to cause expectoration [1], is a common cause. Because the exact definition of the words “chronic” or “recurrent” were decided arbitrarily, Fletcher in 1959 [2] suggested that they should imply that the symptoms have been present on most days for at least 3 months for at least 2 successive winters, a definition that remains the gold standard [3].

Chronic bronchitis (CB) is commonly associated with smoking-induced chronic obstructive pulmonary disease (COPD), and CB symptoms (cough and sputum production) are regularly considered cardinal symptoms of COPD [1]. However, this concept needs clarification since, although CB and COPD can be both caused by smoking, the two entities have very likely dissimilar predisposing backgrounds, as described by Fletcher and Peto [4].

The individual susceptibility to CB in smokers is markedly variable with a higher prevalence in smokers with COPD, affecting 14 to 52% of all patients [5,6,7], but also in smokers without COPD, affecting 20 to 30% [8,9]. These data indicate that smokers could be susceptible to: (1) only CB, as seen in smokers with CB and normal spirometry (noCOPD), also called non obstructive bronchitis; (2) COPD without CB; or (3) both entities, as seen in smokers presenting with COPD-airflow obstruction and dyspnoea-and CB-cough and sputum production. 

Cigarette smoking is the most important known risk factor for the development of chronic bronchitis, however CB is also common in the general population, with prevalence estimates varying widely from 3 to 22% of non-smoking adults, variability due in part to geographical location [6,10,11]. Occupational and biomass exposures, air pollution, long-term exposure to multiple inhaled smokes vapors and particles, asthma, allergies, and bronchiectasis are some of the known factors contributing to the development of chronic bronchitis in the general population [12].

In smokers with established COPD, the presence of chronic bronchitis is associated with an increased lung function decline and increased risk of hospitalization or death [5,13]. However the effects of mucus hypersecretion in the clinical outcomes and the development of chronic airflow obstruction in smokers without COPD is still debatable [14,15]. In order to clarify this issue, it would be important to validate with a longitudinal analysis the relationship between chronic bronchitis and clinical relevant outcomes in smokers and, in particular, in those without airway obstruction.

In an attempt to clarify these relationships, we studied a group of smokers with and without COPD, followed long-term in a single academic institution, who had regularly measured clinical and functional outcomes, in order to assess the possible effects of having chronic bronchitis in smokers without COPD and compare them with the effects in smokers with COPD. 

## 2. Materials and Methods

### 2.1. Study Population

Participants were recruited among smokers who first attended the Pulmonary Clinic at the Hospital Universitario Miguel Servet (Zaragoza, Spain) requesting to be included in a smoking cessation program or referred by other doctors to assess their respiratory health between October 2010 and April 2014. CONSORT diagram with the inclusion criteria are detailed in Appendix A.

At baseline, all subjects were clinically stable, free of major comorbidities, not having had any exacerbations for at least 8 weeks. Subjects with asthma or history of asthma, bronchiectasis, autoimmune diseases, other respiratory diseases, or coexisting malignancy at recruitment were excluded. All subjects underwent a comprehensive clinical and functional examination including spirometry, measurement of transfer factor of the lung for carbon monoxide (DLco), using the European CECA as predicted values [16]. The 6-min walking test (6MWT), the modified Medical Research Council (mMRC) Dyspnoea Scale, and the COPD Assessment Test (CAT) [17] were also conducted. 

COPD was defined by forced expiratory volume in the first second/forced vital capacity (FEV_1_/FVC) <0.7 [3] and smokers without COPD (noCOPD) by FEV_1_/FVC > 0.7 post-bronchodilator (post-bd). Chronic bronchitis was defined as the presence of cough and sputum production for at least 3 months in each of two consecutive years [3]. The annual frequency and type of exacerbations as defined by recent guidelines [3] were collected (as detailed in the online Appendix A).

The baseline visit spirometry was compared to a spirometry performed after 10 ± 4 years of follow-up to assess the functional decline over time and the potential development of COPD. The incidence of comorbidities, including cancer, were prospectively recorded; all-cause mortality and cause of death were recorded up to October 2018.

The study was approved by the human-research review board (IRB.12/2010), and all patients provided informed written consent before any procedure was done.

### 2.2. Statistical Analysis

Patients′ characteristics were described using mean ± SD for continuous variables and using counts and percentages for categorical variables. For continuous variables, normal distributions were tested using the Shapiro–Wilk test. Comparisons among groups were evaluated with Mann–Whitney U tests. Distributions of categorical variables were compared with the χ^2^-test. A zero-inflated Poisson model to assess the total number of exacerbations during the follow-up, to account for the over-dispersion and the relatively high number of zeros in the data set, was used. Analyses of survival probability were performed by Kaplan– Meier survival curves, and differences between survival curves were assessed using the log-rank test. Hazard ratios (HRs) and 95% confidence intervals (CI) were estimated for the variable entered in the final model. All analyses were performed using SPSS (version 25.0.0.1, IBM, Armonk, NY, USA). Statistical significance was assumed for a *p* value < 0.05.

## 3. Results

Among the 511 smokers included in the study, 302 (59%) had COPD (COPD) and 209 (41%) did not have COPD (noCOPD). COPD were older (62 ± 8 vs. 52 ± 11 years; *p* < 0.001), smoked more (49 ± 25 vs. 35 ± 19 packs-years; *p* < 0.001), had lower FEV_1_ post-bd % pred (68 ± 19 vs. 95 ± 15% pred; *p* < 0.001), and a higher prevalence of chronic bronchitis (CB) (137/302, 45% vs. 56/209, 27%; *p* < 0.01) than noCOPD. The percentages of males and females were not different between subjects with and without chronic bronchitis (CB). In particular, in noCOPD with CB the percentage of males was 71% and in those without CB it was 70%, whereas in COPD with CB it was 91% and in those without CB it was 90% (Table 1 and Table 2).

There was no difference in the treatment modalities and type of treatment (single, double or triple inhaled medications) in the COPD patients with or without CB, thus, it is very unlikely that the treatment modality influenced outcomes (Appendix A).

### 3.1. Smoking Habits and Prevalence of Chronic Bronchitis

In the noCOPD population, 11% without CB and 5% with CB quit smoking, whereas in those with COPD, 15% without and 5% with CB quit smoking. The prevalence of CB in noCOPD smokers was 27% and in smokers with COPD was 45%, and this prevalence was maintained during the follow-up despite the changes in smoking habits. The percentage of subjects with CB was similar in all GOLD stages (49% GOLD 1; 43% GOLD 2 and 45% GOLD 3–4; *p* = 0.1) and significantly different from the percentage in smokers noCOPD (27%; *p* = 0.01).

### 3.2. Impact of Chronic Bronchitis on Lung Function

NoCOPD smokers with CB had a lower FEV_1_ post-bd % pred (90 ± 14 vs. 97 ± 15%, *p* < 0.01) and a lower DLco % pred (81 ± 28 vs. 87 ± 17 %; *p* = 0.01) (even if both within normal limits) than the noCOPD without CB (Table 1, Figure 1A,B). Mild expiratory flow at 25–75% of FVC (MEF 25–75% pred) was significantly lower in noCOPD with CB than in the noCOPD without CB (70 ± 21 vs. 77 ± 21 % pred, *p* < 0.01) (Table 1), indicating a dysfunction at the level of the small airways either due to remodelling or mucus plugging, or both. Having CB did not influence the degree of abnormality of FEV_1_ and DLco in smokers with COPD, suggesting that in COPD this effect was masked by the functional abnormalities that characterize the disease (Table 2 and Figure 1A,B).

### 3.3. Impact of Chronic Bronchitis on Clinical Performance

noCOPD with CB subjects were significantly more dyspnoeic, as measured by the % of subjects with MRC ≥ 2 (54 vs. 27%; *p* < 0.01), and performed worse in the 6-min walking test (452 ± 120 vs. 490 ± 111 m; *p*= 0.05) than noCOPD without CB (Table 1). Similarly, smokers with COPD and CB were significantly more dyspnoeic than those without CB (55 vs. 41%; *p* < 0.001), although they performed similarly in the 6 min walking test (367 ± 123 vs. 383 ± 95 m; *p* = 0.55, Table 2).

The percentage of smokers with an abnormal COPD Assessment Test (CAT), a questionnaire designed to measure the impact of COPD on a person′s life (>10 is abnormal), was higher in smokers with chronic bronchitis than in those without in both subjects with or without COPD. This is not a surprising finding, since 10 points out of 40 maximum points are assigned to cough and sputum production in the CAT score (Table 1 and Table 2 and Figure 2).

### 3.4. Impact of Chronic Bronchitis on Exacerbations

In noCOPD smokers, the presence of chronic bronchitis doubles the number of exacerbations (“flare-ups”) per year when compared to noCOPD without CB (0.79 ± 0.013 vs. 0.38 ± 0.002 exacerbations/year; *p*= 0.007), a number similar to the number of exacerbations present in COPD subjects without CB. The effects of chronic bronchitis in smokers with COPD had a similar, but not significant trend (1.19 ± 0.008 vs. 0.84 ± 0.005 exacerbations/year; *p* = 0.09, Table 1 and Table 2, and Figure 3).

### 3.5. Impact of Chronic Bronchitis on FEV_1_ Decline

In noCOPD smokers, the presence of CB had no effect in the FEV_1_ decline measured over 10 ± 4 years (32 ± 34 mL/year in noCOPD without CB vs. 36 ± 44 mL/year in noCOPD with CB; *p* = 0.47, Table 1). During follow-up, 14% of smokers with noCOPD developed COPD, of which 63% did not have CB and 37% had it, indicating that in this population, having CB does not affect the FEV_1_ decline neither is it a risk factor for the eventual development of COPD. Contrary to the noCOPD, the FEV_1_ decline in COPD smokers was dependent in part on the presence of CB, since COPD with CB decline more than COPD without CB (41 ± 48 vs. 22 ± 53 mL/year; *p* < 0.01, Table 2), and in part on the smoking activity, which further accelerates the FEV_1_ decline [18].

### 3.6. Impact of Chronic Bronchitis on Survival

During follow-up, 134 out of 512 smokers died; 72 (54%) with chronic bronchitis and 62 (46%) without chronic bronchitis. CB was associated with decreased survival both in the noCOPD (HR 2.97; 95%CI 1.17–7.48; *p* < 0.01) and COPD smokers (HR 2.61; 95%CI 1.12–2.36; *p* < 0.01) (Figure 4).

## 4. Discussion

Chronic cough and sputum production are among the most commonly reported medical symptoms [2]. Cough and sputum expectoration for most days for at least 3 months during at least 2 consecutive years are considered the epidemiological criteria of chronic bronchitis (CB) [3]. However, productive cough without fulfilling the two years interval included in the definition of chronic bronchitis is also used in some studies or as a clinical guide [19].

The prevalence of CB symptoms in individuals aged 40 years or older ranges from 3% to 22% [6,10,11]; the variability is due in part to the geographical location [6] or living in rural areas [7]. The prevalence of CB increases among smokers (38% in our study had CB) and with the presence of COPD (45% of our COPD population had CB, whereas only 27% of the noCOPD had it), in line with previous reports [6,7,8,9,10,11,12].

There is extensive literature describing the effects of CB in smokers with COPD, however the effects of what was called “non obstructive bronchitis” (that is bronchitis with no airflow limitation) in the clinical outcomes of these smokers is much less well defined [9]. The evaluation of our “real life” population of smokers without COPD prospectively followed for about 10 years, showed that 27% of the noCOPD smokers had CB, and that the presence of CB significantly influenced symptoms as well as survival. These patients had worse lung function (lower FEV_1_ and DLco, even if within normal limits), more exercise limitation, were more dyspnoeic, had more exacerbations, and a higher mortality rate than noCOPD without CB. These findings demonstrate the important morbidity that accompanies the presence of “non obstructive bronchitis”, morbidity that is preventable and treatable, since in most cases chronic bronchitis is not necessarily a chronic condition but can vary over time, with the main determinant of this variation being smoking status. Persistently or resuming smoking cigarettes increases the odds of having or developing chronic bronchitis, and quitting smoking increases the odds of the resolution of chronic bronchitis [19].

The “non-obstructive chronic bronchitis” was initially described in Europe and differentiated from obstructive CB [4]. On the other hand, CB was not recognized in the US as an entity, and smokers with airflow obstruction were reported as having emphysema, which created a problematic situation eventually solved by the unifying name of chronic obstructive pulmonary disease (COPD). After the term COPD became popular, chronic bronchitis, originally recognized as a stand-alone condition [4], was commonly yet incorrectly viewed solely as a COPD phenotype and hence rarely diagnosed in individuals without COPD [1]. Thus, the diagnosis of COPD in a non-obstructive CB would be a false diagnosis of COPD that unfortunately is found in a high percentage of patients [20]. Hence, a smoker with CB ought to have a spirometry documenting airflow obstruction before the diagnosis of COPD could be made.

Consequently, chronic bronchitis should be considered as a separate entity that can exist even without airflow obstruction (COPD). Indeed, in our study, chronic bronchitis was present in 27% of the non-COPD smokers, with some discordance with the few data on the prevalence of chronic bronchitis in smokers without COPD, reporting less than 10% [10,13,21,22].

We found that chronic bronchitis has a significant impact on lung function and symptoms in noCOPD smokers, in whom FEV_1_ and DLco are significantly reduced (even if well within the normal limits), when compared with noCOPD without CB. Of interest, CB does not have the same functional effects in smokers with COPD, probably because the pathological abnormalities, including the presence of emphysema, might overpower the added effects of CB. This finding confirms the results of the large cross-sectional study by Kim and coworkers [23] but is in disagreement with other observations reporting worse lung function in COPD subjects with chronic bronchitis [13].

In our cohort, for similar values of FEV_1_, noCOPD smokers with CB had lower exercise capacity than those without CB, and the presence of CB was related to the degree of dyspnoea both in the COPD and noCOPD groups when compared with those without CB, a finding underlining the poor relationship often seen between dyspnoea and degree of airflow limitation. An explanation for this symptom probably resides in the extra amount of ventilation necessary for any degree of work even in noCOPD smokers, as demonstrated in a recent report [24]. These smokers might have early airway abnormalities and abnormal DLco, secondary to gas exchange abnormalities aggravated by the presence of CB, explaining the presenting symptoms [18]. Interestingly, the percentage of patients with a severity of dyspnoea above mMRC ≥ 2 was similar in chronic bronchitis with and without COPD, probably due to the fact that smokers noCOPD were able to exercise more than smokers with COPD, thus experimenting similar degrees of dyspnoea for a heavier exercise load. Similarly, it has been reported that in smokers the presence of chronic bronchitis is associated with worse symptoms and quality of life than the presence of COPD [25]. This may explain why a wrong diagnosis of COPD is common in people who report symptoms compatible with chronic bronchitis [20] without airflow limitation or normal spirometry. Thus, such patients ought to always have a spirometry before the COPD diagnosis could be made.

In our population, the presence of chronic bronchitis was associated with a higher frequency of exacerbations or “flare ups” of their symptoms, including increased volume of sputum, often colored, at time haemoptysis in the noCOPD group. These events should be taken into consideration in smokers without airway obstruction, because they impact on the quality of life and work capacity [26,27]. Conversely we found no significant association between chronic bronchitis and exacerbation rate in the COPD group, confirming previous data [28,29] but conflicting with other studies [23,30,31,32] showing an association between presence of chronic bronchitis and increased rate of exacerbations in COPD subjects.

The contribution of CB to the clinical presentation of smokers, especially in those with noCOPD, could be better understood by considering chronic bronchitis not only as increased sputum production from bronchial glands in the large airways that would be expectorated, but as part of the so called “muco-obstructive” disease [33]. This disease is characterized by abnormally raised MUC5AC mucin concentrations [34], increased sputum production, and mucus hyperconcentration throughout the bronchial tree, responsible for the abnormal mucus transport and intrapulmonary mucus accumulation, the basis to the pathogenesis of chronic bronchitis [33,35]. The accumulation of mucus difficult to expectorate would form mucus plugs within airway lumens, which would favor the development of airflow obstruction, inflammation, and intermittent infection [35,36]. Luminal plugging in peripheral airways, identified by CT scans, is a frequent finding significantly associated to CB, which suggests that mucus plugging may play an important role in the pathophysiology of airflow obstruction in smokers with CB [37] and could help understanding the clinical effects and outcomes seen in these subjects.

The development of CB in smokers without COPD has been considered an early marker of susceptibility to the effect of cigarette smoking, able to identify a subgroup of patients with an increased risk of developing COPD [13,38], especially during middle age [7,39]. However, the clinical utility of the subgroups identified in any cross-sectional analysis would need to be validated longitudinally against clinically relevant outcomes [29]. In our population of smokers without COPD with a mean age of 52 years followed for 10 years, 14% developed COPD with no difference between those with and without CB (Table 1), suggesting that CB was not a clear risk or warning for the development of COPD during the long longitudinal follow-up.

Nevertheless, chronic bronchitis impacts on FEV1 decline, but only in smokers with COPD in our study, which suggests that chronic bronchitis is probably a factor that aggravates an already existing COPD, but that is not sufficient to impact on lung function decline in smokers with normal lung function. In line with our findings, a population-based cohort study [31] confirmed the excess lung function decline in COPD with chronic bronchitis.

An important finding in our population is that smokers without COPD but with CB have a significant increase in all-cause mortality when compared with noCOPD smokers without CB, a fact that has been well known for smokers with COPD and CB, as we could demonstrate in our population. However, it is only more recently that it has been shown that even noCOPD smokers with CB have a lower survival rate [40]. Interestingly enough, this is not the case with the chronic bronchitis seen in non-smokers [41].

Yet, the link between non-obstructive CB and mortality may not necessarily be a causal relationship, since heavy smoking exposure is also a risk factor for other causes of death like cardiovascular death and lung cancer.

We acknowledge that being a single-center study based in an ambulatory setting has a potential bias selection. In this setting, smokers wanting to quit and those being referred for care are likely more symptomatic and might not be representative of the general population. Nonetheless, having a population of smokers without COPD and a mean age of 53 years, along with a group with COPD, carefully characterized and followed longitudinally by the same group of physicians adds validity to the study.

In clinical practice, repeated evaluation for the presence or absence of chronic bronchitis could have great implications for the individual patient′s prognosis, including suspicion for new comorbid conditions or complications. Certainly, the inconsistency (recurrence and/or resolution) [42] of chronic bronchitis should not be contextualized simply as “a normal part of having COPD,” as perhaps has been done historically.

## 5. Conclusions

In conclusion, a smoker presenting with symptoms of chronic bronchitis, cough, and sputum production for more than 2 years, might have non-obstructive bronchitis or might have COPD and chronic bronchitis when the spirometry is abnormal. In both cases, the presence of chronic bronchitis very significantly will affect symptoms, quality of life, and survival, underlining the importance of recognizing the condition and managing it accordingly.

## Figures and Tables

**Figure 1 jcm-11-04886-f001:**
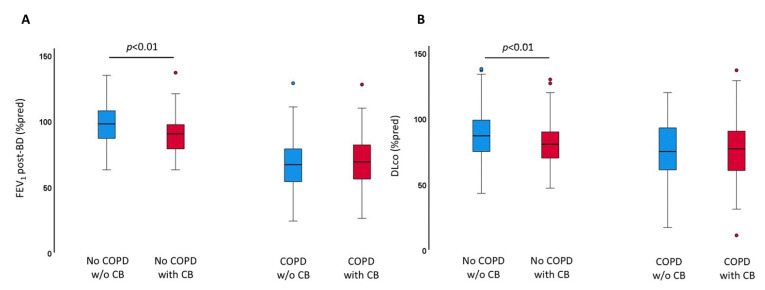
(**A,B**). Effects of Chronic Bronchitis (CB) on FEV_1_ and diffusion capacity (DLco). (**A**) CB worsens the FEV_1_ (%pred) (*p* < 0.01) and (**B**) DLco (%pred) (*p* < 0.01) in smokers with no COPD, but not in smokers with COPD. Bottom and top of each box plot: 25th and 75th percentiles; solid line: median; brackets: 10th and 90th percentiles. CB: Chronic Bronchitis; COPD: Chronic Obstructive Pulmonary Disease; FEV_1_: forced expiratory volume in the first second; DLco: diffusing capacity of the lung for carbon monoxide.

**Figure 2 jcm-11-04886-f002:**
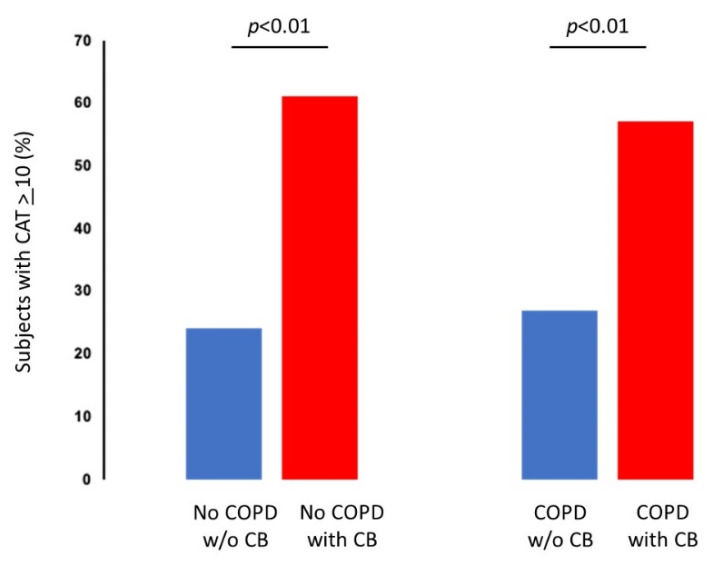
Effects of Chronic Bronchitis (CB) on the COPD Assessment Test (CAT score). The presence of CB increased the percentage of subjects with abnormal CAT score (CAT ≥ 10) both in smokers with and without COPD. COPD: Chronic Obstructive Pulmonary Disease; CAT: COPD Assessment Test.

**Figure 3 jcm-11-04886-f003:**
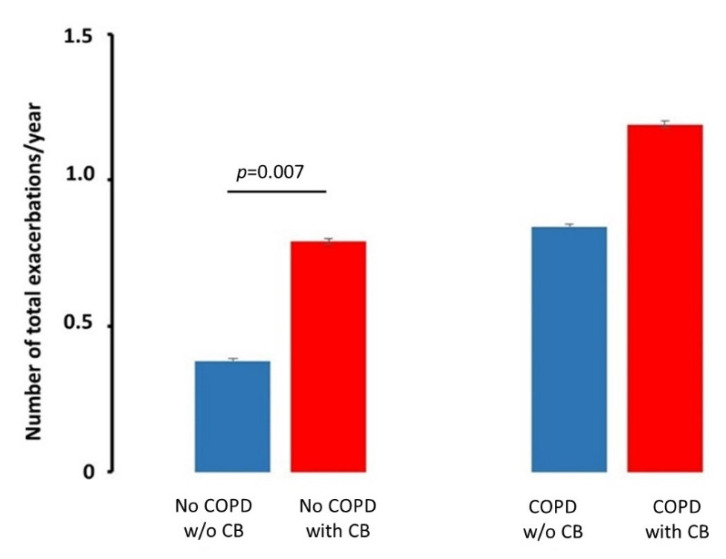
Effects of Chronic Bronchitis (CB) on the number of total exacerbations per year. The presence of CB worsens the total number of exacerbations in smokers without COPD but not in those with COPD. Histograms represent mean ± SD. COPD: Chronic Obstructive Pulmonary Disease.

**Figure 4 jcm-11-04886-f004:**
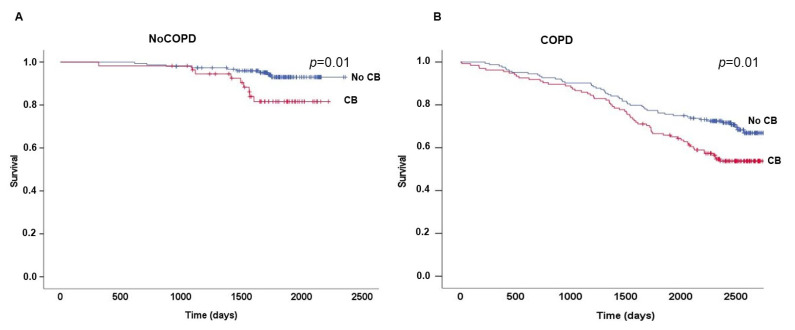
(A,B) Effects of Chronic Bronchitis (CB) on survival. Kaplan–Meier plots showing that the presence of CB decreases survival in (**A**) smokers without COPD (*p* = 0.01) and in (**B**) those with COPD (*p* = 0.01). COPD: Chronic Obstructive Pulmonary Disease.

**Table 1 jcm-11-04886-t001:** Clinical characteristics of subjects without COPD (noCOPD).

	NoCOPD all Population *n* = 209	NoCOPD w/o CB *n* = 153 (73%)	NoCOPD with CB *n* = 56 (27%)	*p*
Age (years)	52 ± 11	51 ± 11	53 ± 12	0.43
Males *n* (%)	147 (70%)	107 (70%)	40 (71%)	0.63
Smoking history (pack years)	35 ± 19	34 ± 18	38 ± 21	0.21
Active smokers *n* (%)	108 (51%)	76 (49%)	32 (57%)	0.31
FEV_1_ post-bd (%pred.)	95 ± 15	97 ± 15	90 ± 14	<0.01
post-bd FEV_1_/FVC (%)	78 ± 5	79 ± 5	78 ± 5	0.10
Decline of FEV_1_ per year (ml/year)	33 ± 37	32 ± 34	36 ± 44	0.47
MEF 25–75 post-bd (% pred.)	75 ± 21	77 ± 21	70 ± 21	<0.01
DLco (% pred)	80 ± 21	87 ± 17	81 ± 28	0.01
Subjects with mMRC ≥ 2, *n* (%)	71 (34%)	41 (27%)	30 (54%)	<0.01
Subjects with CAT ≥ 10, *n* (%)	70 (33%)	36 (24%)	34 (61%)	<0.01
Distance at 6-min walking test (m)	481 ± 114	490 ± 111	452 ± 120	0.05
Subjects with at least one exacerbation, *n* (%)	73 (35%)	46 (30%)	27 (48%)	0.01
Number of total exacerbations/year	0.49 ± 0.007	0.38 ± 0.002	0.79 ± 0.013	0.007
Subjects who develop COPD at follow-up *n* (%)	30 (14%)	19 (12%)	11 (20%)	0.1

Data are presented as number (%) or mean ± SD. *p* value refers to Mann–Whitney test or χ2 test. w/o CB: without Chronic Bronchitis; CB: Chronic Bronchitis; COPD: Chronic Obstructive Pulmonary Disease; FEV_1_: forced expiratory volume in the first second; FVC: forced vital capacity; MEF: maximum mid expiratory flow at 25–75% of FVC; DLco: diffusing capacity of the lung for carbon monoxide; mMRC: modified Medical Research Council; CAT: COPD Assessment Test.

**Table 2 jcm-11-04886-t002:** Clinical characteristics of subjects with COPD.

	COPD all Population *n* = 302	COPD w/o CB *n* = 165 (55%)	COPD with CB *n* = 137 (45%)	*p*
Age (years)	62 ± 8	61 ± 8	63 ± 8	0.08
Males *n* (%)	276 (91%)	150 (90%)	126 (91%)	0.53
Smoking history (pack years)	49 ± 25	50 ± 26	49 ± 25	0.79
Active smokers *n* (%)	103 (34%)	53 (32%)	50 (36%)	0.4
FEV_1_ post-bd (% pred.)	68 ± 19	67 ± 19	69 ± 20	0.38
post-bd FEV_1_/FVC (%)	54 ± 11	54 ± 11	54 ± 12	0.73
GOLD 1, *n* (%)	81 (27%)	41 (25%)	40 (29%)	0.65
GOLD 2, *n* (%)	169 (56%)	96 (58%)	73 (53%)	
GOLD 3–4, *n* (%)	53 (17%)	28 (17%)	24 (18%)	
Decline of FEV_1_ per year (mL/year)	31 ± 51	22 ± 53	41 ± 48	<0.01
MEF 25–75 post-bd (% pred.)	27 ± 13	27 ± 13	28 ± 14	0.40
DLco (% pred)	80 ± 21	76 ± 21	77 ± 22	0.32
Subjects with mMRC ≥ 2, *n* (%)	144 (47%)	68 (41%)	76 (55%)	<0.01
Subjects with CAT ≥ 10, *n* (%)	123 (41%)	45 (27%)	78 (57%)	<0.01
Distance at 6 min walking test (m)	376 ± 109	383 ± 95	367 ± 123	0.55
Subjects with at least one exacerbation, *n* (%)	162 (53%)	84 (51%)	78 (57%)	0.16
Number of total exacerbations/year	1.00 ± 0.007	0.84 ± 0.005	1.19 ± 0.008	0.09

Data are presented as number (%) or mean ± SD. *p* value refers to Mann–Whitney test or χ2 test. w/o CB: without Chronic Bronchitis; CB: Chronic Bronchitis. COPD: Chronic Obstructive Pulmonary Disease; FEV_1_: forced expiratory volume in the first second; FVC: forced vital capacity; MEF: maximum mid expiratory flow at 25–75% of FVC; DLco: diffusing capacity of the lung for carbon monoxide; mMRC: modified Medical Research Council; CAT: COPD Assessment Test.

## Data Availability

Not applicable.

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
