# Peer review of "Chronic Bronchitis Affects Outcomes in Smokers without Chronic Obstructive Pulmonary Disease (COPD)"

_jcm, 2022, doi:10.3390/jcm11164886_

Round 1

Reviewer 1 Report

The manuscript reports the effect of chronic bronchitis designation in a cohort of people who smoked, followed over time at a single center. The manuscript contributes to the growing knowledge in the literature suggesting that the presence of CB is associated with clinically significant symptoms and reduced quality-of-life in the absence of spirometrically-defined COPD. The longitudinal nature of the study (10 +/- 4 years of follow-up) adds to the value of the analyses.  The literature is appropriately cited for this complex topic.

Major Comments:

A major demographic feature is missing from the tables, i.e., sex. If sex percentage is different among compared cohorts, models should be adjusted for sex.

Similarly, current smoking status, sex, and age are often included in the models to seek hidden effects that might lead to misinterpretation. While this adds complexity, it is often a more rigorous approach.  Did the participants change their smoking status across time?

Exacerbation data is often modeled in zero-inflation models due to the fact that many people do not experience exacerbation.  It is not clear what the distribution of exacerbations was or how it was exactly calculated.  Can you provide more detail about exacerbations?

The bar graphs describing mMRC and CAT for example, may not be necessary since the data is in the tables. 

It would be valuable to know how many of the participants who report CB at baseline continue to report CB at the end of the study.

It might be useful to consider a table describing in more detail the cohort at the "end" of the 10 years.

The limitations should be expanded if appropriate to include complexity of changing smoking status across time, potential bias due to patient selection in this setting (smokers wanting to quit, smokers being referred) who might start with more symptoms than the general population. In addition, it is not clear if any of the COPD treatments that some patients would be prescribed/use would affect the findings. 

Minor:

Line 27... ambulatory setting?

It may be worth it at some point to mention that the spirometry-defined functional loss might be mostly explained by emphysema in COPD.

Figure 1.  Bar graphs are sort of going out of style, with many journals now wishing to see all the data points and distributions (box and whisker and variations on that style may be more informative).

Author Response

Major Comments:

Q1R1: A major demographic feature is missing from the tables, i.e., sex. If sex percentage is different among compared cohorts, models should be adjusted for sex.

A1R1: Sex information has been added to Tables 1 and 2. In our cohorts study, the percentages of males and females are not different between subjects with and without chronic bronchitis (CB). In particular, in noCOPD with CB the percentage of males was 71% and in those without CB it was 70%, while in COPD with CB it was 91% and in those without CB it was 90%. We added the new information about the gender distribution in Tables 1 and 2 and in Results (page 3 lines 130-134).

Q2R1: Similarly, current smoking status, sex, and age are often included in the models to seek hidden effects that might lead to misinterpretation. While this adds complexity, it is often a more rigorous approach. Did the participants change their smoking status across time?

A2R1: Age was not different between subjects with and without chronic bronchitis (53±12 vs 51±11 yrs in the noCOPD cohort and 63±8 vs 61±8 in the COPD cohort). A small proportion of patients changed their smoking status during the follow-up period. Among the noCOPD subjects, 11% of patients without CB and 5% with CB quitted smoking. Among subjects with COPD, 16% of those without CB and 5% of those with CB quitted smoking. These percentages were not significantly different by χ2 test (p=0.12). These data are now included in the Results (page 3 lines 140-141).

Q3R1: Exacerbation data is often modeled in zero-inflation models due to the fact that many people do not experience exacerbation. It is not clear what the distribution of exacerbations was or how it was exactly calculated. Can you provide more detail about exacerbations?

A3R1: We agree with the reviewer that many people did not experience exacerbations. In fact, a high proportion of subjects in the study did not exacerbate during the time of observation. As suggested by the reviewer, we used a zero-inflated Poisson model to assess the total number of exacerbations during the follow up, to account for the over-dispersion and the relatively high number of zeros in the data set (Methods page 3, lines 117-119). The corrected data, in which only the standard deviation but not the mean changed, are now reported in the Tables 1 and 2. In our study the annual exacerbation rate of each patient was calculated by dividing the number of exacerbations by the number of days they participated in the study and multiplying by 365. In this calculation we also included the patients who never exacerbated during the study (that is, even patients who have zero exacerbations). To clarify this point we have plotted the distribution of number of exacerbations/year in the online supplementary file (Figure E2), and added more details about the calculation of exacerbations in the online supplementary file.

Q4R1: The bar graphs describing mMRC and CAT for example, may not be necessary since the data is in the tables.

A4R1: We are aware of the duplication of data having the graphs and tables. We have maintained the tabular data and removed the graph of the mMRC, but we would like to keep the one on the CAT. The reason for this is that it is not clearly realized how much the CAT, a score that is supposed to assess the overall clinical status of a patient with COPD, depends on having or not chronic bronchitis. The graph makes this point fairly obvious.

Q5R1: It would be valuable to know how many of the participants who report CB at baseline continue to report CB at the end of the study.

A5R1: In our study, patients reporting CB symptoms at baseline continue to report CB during follow-up in both COPD and noCOPD groups. In this ambulatory setting, smokers wanting to quit and those being referred for care are likely more symptomatic and might not be representative of a general population (Discussion, page 9 lines 330-333).

Q6R1: It might be useful to consider a table describing in more detail the cohort at the "end" of the 10 years.

A6R1:  At the last visit:

1.- In noCOPD age was 61±10 years (was 52±11 at the first visit; p=0.01); in COPD it was 70±8 (was 62±8 at the first visit; p=0.01).

2.- In noCOPD the pack years of smoking was 42±24 (was 35±19 at the first visit; p=0.01); in COPD it was 52±32 (was 49±25 at the first visit; p=0.01).

3.- In noCOPD 11% w/o CB and 5% with CB quitted smoking. In COPD 15% w/o CB and 5% with CB quitted smoking. Added to page 3 lines 140-141 of the results section.

4.- In noCOPD FEV1 % pred was 88±17 (was 95±15 at first visit; p=0.01) and the FEV1/FVC % 76±7 (was 78±5 at first visit; p=0.01). In COPD FEV1 % pred was 62±20 (was 68±19 at first visit; p=0.01) and the FEV1/FVC %: 54±13 (was 54±11 at first visit; p=0.41).

5.- As mentioned in the response to question 5 patients reporting CB symptoms at baseline continue to report CB during follow-up in both COPD and noCOPD groups.

Q7R1: The limitations should be expanded if appropriate to include complexity of changing smoking status across time, potential bias due to patient selection in this setting (smokers wanting to quit, smokers being referred) who might start with more symptoms than the general population. In addition, it is not clear if any of the COPD treatments that some patients would be prescribed/use would affect the findings.

A7R1: As suggested the limitations paragraph has been expanded to include the very pertinent reviewer’s suggestions (Discussion, page 9, lines 330-335).

Also a table showing the therapy in COPD subjects with and without CB (with no statistical differences in the treatment modalities) has been added to the supplementary file (Table E1) and commented in the Results page 3 lines 135-138.

Therapy

COPD

(n=302)

COPD w/o CB

(n=165)

COPD with CB

(n=137)

p

ICS

179/302 (80%)

96/165 (58%)

83/137 (60%)

NS

LAMA

35/302 (11%)

20/165 (12%)

15/137 (11%)

NS

LABA/LAMA

10/302 (3%9)

6/165 (4%)

4/137 (3%)

NS

LAMA/ICS

52/302 (17%)

31/165 (19%)

21/137 (15%)

NS

TripleTherapy

120/302 (40%)

59/165 (36%)

61/137 (44%)

NS

Minor:

Q1R1: Line 27... ambulatory setting?

A1R1: It's correct. We changed the sentence according to the reviewer’s suggestion.

Q2R1: It may be worth it at some point to mention that the spirometry-defined functional loss might be mostly explained by emphysema in COPD.

A2R1: We thank the reviewer for the suggestion. The sentence “including the presence of emphysema” has been added to the paragraph in Discussion, page 9, lines 257-260.

Q3R1: Figure 1.  Bar graphs are sort of going out of style, with many journals now wishing to see all the data points and distributions (box and whisker and variations on that style may be more informative).

A3R1: We changed the figure 1 according to the reviewer’s suggestion.

Reviewer 2 Report

The title of the article fully reflects the content of the article.

The abstract summarizes the information necessary for the reader: the relevance of the research, the purpose, methods, results. The conclusion is concrete, logically follows from the results.

The presented keywords are necessary and reflect the research topic presented by the authors.

In the "Introduction" section, the authors describe chronic bronchitis and the epidemiology of the disease quite fully. It is shown that it is necessary to clarify the concept of the association of chronic bronchitis with smoking-induced chronic obstructive pulmonary disease (COPD), that the symptoms of chronic bronchitis are regularly considered cardinal symptoms of COPD. The effect of mucus hypersecretion on clinical outcomes and the development of chronic airway obstruction in smokers without COPD is still debatable. The relevance of the problem is presented correctly and clearly. The aim of the study was to study the possible effect of chronic bronchitis on clinical outcomes in smokers without COPD (noCOPD) and compare them with the effects in smokers with COPD (COPD).

The section "Materials and methods" indicates that the study participants were recruited among smokers. The study was conducted on the basis of the pulmonology clinic at the Universitario Miguel Servet Hospital (Zaragoza, Spain). The terms of observation of patients are presented. Information on the comprehensive clinical and functional examination of patients and basic examinations of patients for the determination of COPD and chronic bronchitis is provided to the required extent. It is important for the authors to register the frequency of concomitant diseases and the type of exacerbations, mortality from all causes and causes of death. The study was approved by the Human Research Supervisory Board (IRB.12/2010), and all patients gave informed written consent prior to any procedure. The design of the study is clear. The presented statistical analysis of the data corresponds to the task of this study.

The section "Results" presents the number of patients and groups of patients. The results presented by the authors are important and necessary for pulmonology, reliable, and give progress in understanding the course of chronic bronchitis in patients of various groups.

With the involvement of the literature in the "Discussion" section, the authors discussed the results well.

The authors' conclusion is relevant and fully consistent with the results obtained, is that the presence of chronic bronchitis in smokers without COPD will significantly affect symptoms, quality of life and survival, which are known to be significantly improved by smoking session.

All tables are clear and legible, and are necessary to understand the results of the study. The drawings complement the article.

The article does not cause any concerns. The manuscript did not cause any ethical problems. Statistical analysis corresponds to the study. All references to publications presented by the authors in the article are necessary and correct, made in the right style. Of the 42 links that are presented in the article, 18 links are from the last 5 years (2017-2022). I have no concerns about the similarity of this article with other articles published by the same authors.

Competing interests of authors do not create bias in the presentation of results and conclusions.

Author Response

We thank the reviewer for the positive comments and for appreciating our manuscript.

Reviewer 3 Report

Authors have demonstrated that presence of chronic bronchitis in smokers with noCOPD significantly affect symptoms, quality of life and survival. My comments: Authors have not discussed about tobacco cessation, however writing in the conclusion part. A schematic representation of work-flow (CONSORT Flow chart) can be added. 

Author Response

Q1R2: Authors have not discussed about tobacco cessation, however writing in the conclusion part.

A1R2: We agree with the reviewer and have deleted the sentence both in the abstract (lines 36-39) and the conclusions (page 10, lines 344-349).

Q2R2: A schematic representation of work-flow (CONSORT Flow chart) can be added.

A2R2: We added the CONSORT flow chart in the supplementary file (Figure E1).
